# Using Collaged Data Augmentation to Train Deep Neural Net with Few Data

**Hsun-An Chiang**[*][1]                                                  LABA15454@GMAIL.COM
[1] *LiLee Systems, Inc.*

**Cheng-Shao Chiang**[2]                                    R06922131@CSIE.NTU.EDU.TW
**Chi-Sheng Shih**[2]                                              CSHIH@CSIE.NTU.EDU.TW
[2] *National Taiwan University, Taipei, Taiwan*

**Editors:** Under Review for MIDL 2019

**Introduction**   Object recognition has achieved great breakthrough thanks to the development of Deep Neural Network (DNN). Although many proposed networks can derive scene information on pixel level, they require a large amount of finely annotated data for training. However, datasets for robot-assisted surgery are difficult to acquire. There are many shortcomings when training deep network with limited data. For examples, overfitting and degraded network performance are two significant shortcomings.

In this work, we propose a data augmentation technique, called **collaged data augmentation**(CDA), which synthesizes new training data by the components extracted from original training data. We evaluated the effectiveness of collage approach in the endoscopy surgery scenario, and compared it with other commonly used data augmentation technique. Experiment results show that proposed data augmentation can improve network performance better than other commonly used data augmentation methods. Experiment results show that data augmentation can increase mIoU for up 14.6% for PSPNet and up to 39.83% for DeepLab networks.

**Targeted Problem and System Architecture**   This work aims on the problem of semantic segmentation on endoscopy environment. The problem of semantic segmentation is to, given an image $I \in R^{h \times w \times c}$, predict the class for each pixel $L = C(I), L \in R^{h \times w}$, where L is the per-pixel prediction, $C$ is the classifier, and $m$, $n$, $c$ are height, width and number of channels of the image respectively.

Figure 1 shows the overall system architecture of the proposed method. The core of our system is a classifier for semantic segmentation on images from endoscopy surgery, which is expected to be a deep neural network. Before the classifier is a data augmentation model which will drastically expand the original training set by aggressive data augmentation which will be elaborated later so that our deep neural network classifier can be trained effectively on the original low-volume training set.

**Background and Related Work**   Many works have been proposed for semantic segmentation. Among them, deep neural network approaches show promising improvement in last few years. Chen et. al (Chen et al., 2018) proposed DeepLab which uses Atrous

---

[*] Contributed equally

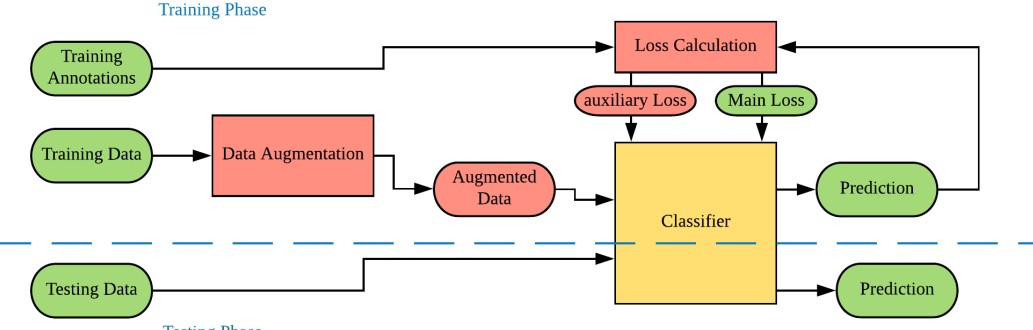

Figure 1: System Architecture of Proposed Method

Spatial Pyramid Pooling (ASPP) to capture features of different scales and receptive fields. Zhao et. al (Zhao et al., 2017) proposed PSPNet which uses pyramid pooling module to conduct global pooling of different scales on the feature map output by a deep network, and then rescale and concatenate results of global pooling with original feature map for final prediction. The method proposed in this paper will be evaluated against these two methods.

The effectiveness of supervised deep neural network methods depend not only on the network model but also on the amount of annotated training data. Data augmentation is a technique to increase the size of training data. The rationale of data augmentation is to synthesize training data by modifying existing data so as to increase the diversity of the training datasets. While synthesizing new data, the fundamental requirement is to maintain the properties of the object on interests. Krizhevsky et. al (Krizhevsky et al., 2012) proposed to randomly cropped, horizontally flipped and altered color intensity of images when training network on ImageNet dataset, which is crucial for their network to reduce overfitting. Simard et. al (Simard et al., 2003) proposed to do elastic transformation on training data when training network on MNIST dataset. Elastic transformation will randomly compute a displacement field, which indicates how each pixel will transform to a new location. Ronneberger et. al (Ronneberger et al., 2015) also used elastic transformation on microscopical image dataset.

**Collage Data Augmentation (CDA) Algorithm**  The proposed data augmentation technique is named *collage data augmentation*, which generates new training data and labels by gluing together pieces that are duplicated from other training samples. The process looks after the process of assemble different materials to a new artifact.

Collage Data Augmentation (CDA) Algorithm is shown in Algorithm 1. The major concept of this algorithm is to stick the objects labeled in different images on to a new image, rather than manipulating the existed objects in the image. One major rule of the CDA algorithm is that the instruments will always be placed on top of the organs. It is because the instruments must be seen by the surgeon so as to operate. The algorithm consists of two iterations. In the first iteration, the algorithm iterates through all label $l$ in organ classes, randomly selects a training image $I_i$ and annotation $L_i$ pair, and copy all pixels of the object with class $l$ in image $I_i$ and label $L_i$ to collaged image $I_c$ and label $L_c$. The second iteration repeats the process for instruments classes.

**Algorithm 1:** Collaged Data Augmentation (CDA) Algorithm

---

**Input:** The training image set, $I_t$; the training annotation set, $L_t$

**Output:** The collaged image, $I_c$ and corresponding annotation $L_c$

**for** *Each $L_j$ in organ classes* **do**
 Randomly choose an image $I_i$ from $I_t$, with it's corresponding annotation $L_i$;
 $I_c(p) = I_i(p)$ and $L_c(p) = L_i(p) \ \forall p \in P$, where $P$ is the set of all pixels in $I_i$ that are in class $l_j$;

**end**

**for** *Each $L_j$ in instrument classes* **do**
 Randomly choose an image $I_i$ from $I_t$, with it's corresponding annotation $L_i$;
 $I_c(p) = I_i(p)$ and $L_c(p) = L_i(p) \ \forall p \in P$, where $P$ is the set of all pixels in $I_i$ that are in class $l_j$;

**end**

return $I_c, L_c$;

---

Figure 2 shows several example images and corresponding semantic labels generated by CDA algorithm.

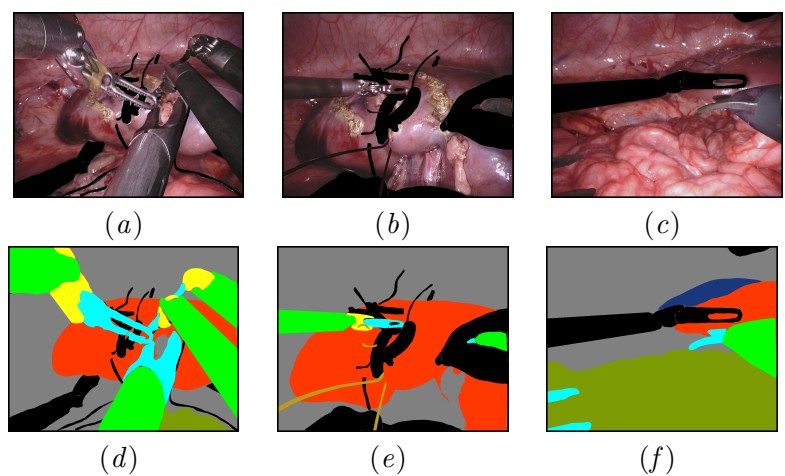

$(a)$           $(b)$           $(c)$

$(d)$           $(e)$           $(f)$

Figure 2: Example result of augmented data by CDA Algorithm

The CDA algorithm is evaluated for PSPNet and DeepLab. Dataset used in the evaluation are received at MICCAI 2018 Robotic Scene Segmentation Sub-Challenge (for Open Medical Image Computing, 2018), which contains 16 images sequences recorded in robotic porcine nephrectomy procedures using da Vinci Xi systems. Experiment results show that data augmentation can increase mIoU for up 14.6% for PSPNet and up to 39.83% for DeepLab networks.

**Conclusion** In this paper, a new data augmentation technique is proposed. The proposed data augmentation randomly samples objects or background from different images and then collage them together. Experiment results showed that collage is more effective than many commonly used data augmentation methods in endoscopy surgery scenario.

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
