# OpenReview forum: "Using Collaged Data Augmentation to Train Deep Neural Net with Few Data"
_MIDL.io/2019/Conference/Abstract — MIDL Abstract 2019_

### Official Review · AnonReviewer1 · 2019-04-29
**sharp performance improvement due to proposed augmentation method**

**Rating:** 3
**Confidence:** 3

**Review:**

The abstract proposes to create "collage" images containing endoscopic surgical tools to augment data and obtain segmentation of said tools in endoscopic videos as obtained during robotic surgery.

the paper presents results that appear to have improved significantly due to the proposed augmentation technique. the "collage" images are therefore realistic enough to allow training of the method and increased performances.

I think this work is interesting to the community and therefore I support acceptance.

---

### Official Review · AnonReviewer2 · 2019-04-29
**Interesting Idea, not clear-written paper**

**Rating:** 3
**Confidence:** 3

**Review:**

The authors propose a data augmentation strategy to synthesize datasets for robot-assisted surgery. The authors also utilize semantic segmentation to delineate regions of interest.

Pros
- The proposed data augmentation method is an interesting idea for robotic surgery semantic segmentation task in endoscopy images.

Cons
- The main idea of the work is not very clear. The authors claim the augmentation technique to be the main contribution in the beginning and then put the emphasize on semantic segmentation. Please write a small abstract to describe the method at the beginning as the other MIDL submissions.
- The authors claim: “ collage is more effective than many commonly used data augmentation methods in endoscopy surgery scenario”, which is not backed up with results. I do not see any results with state of the art augmentation methods.
- Please provide a table of results including: 1-) results with no augmentation; 2-) results with state of the art augmentation (e.g. rotation, cropping, elastic deformation etc.); 3-) results with proposed approach


Minor issues/suggestions
- data augmentation technique →  techniques
- The sentence “The method proposed in this paper will be evaluated against these two methods.” is not accurate. These methods are utilized to together with data augmentation to show the efficiency of the proposed data augmentation. Please correct the sentence.

---

### Decision · Program_Chairs · 2019-05-06
**Acceptance Decision**

Accept